# Characterization of the Gut Microbiota in Urban Thai Individuals Reveals Enterotype-Specific Signature

**DOI:** 10.3390/microorganisms11010136

**Published:** 2023-01-05

**Authors:** Jiramaetha Sinsuebchuea, Prasobsook Paenkaew, Montree Wutthiin, Thatchawanon Nantanaranon, Kiattiyot Laeman, Weerayuth Kittichotirat, Songsak Wattanachaisaereekul, Sudarat Dulsawat, Montira Nopharatana, Namol Vorapreeda, Sakarindr Bhumiratana, Supapon Cheevadhanarak, Sawannee Sutheeworapong

**Affiliations:** 1Bioinformatics and Systems Biology Program, School of Bioresources and Technology, and School of Information Technology, King Mongkut’s University of Technology Thonburi (KMUTT), Bangkok 10150, Thailand; 2Systems Biology and Bioinformatics Unit, Pilot Plant Development and Training Institute, King Mongkut’s University of Technology Thonburi (KMUTT), Bangkok 10150, Thailand; 3Innovation Ecosystem, Knowledge Xchange, King Mongkut’s University of Technology Thonburi (KMUTT), Bangkok 10600, Thailand; 4School of Food Industry, King Mongkut’s Institute of Technology Ladkrabang (KMITL), Bangkok 10520, Thailand; 5Fungal Biotechnology Unit, Pilot Plant Development and Training Institute, King Mongkut’s University of Technology Thonburi (KMUTT), Bangkok 10150, Thailand; 6Department of Food Engineering, Faculty of Engineering, King Mongkut’s University of Technology Thonburi (KMUTT), Bangkok 10140, Thailand; 7Research, Innovation and Partnerships Office, King Mongkut’s University of Technology Thonburi (KMUTT), Bangkok 10140, Thailand; 8School of Bioresources and Technology, King Mongkut’s University of Technology Thonburi (KMUTT), Bangkok 10150, Thailand

**Keywords:** gut microbiome, gut microbiota, enterotype, Ruminococcaceae, baseline gut microbiome, urban, co-occurrence network, 16S rRNA sequencing, Thai

## Abstract

Gut microbiota play vital roles in human health, utilizing indigestible nutrients, producing essential substances, regulating the immune system, and inhibiting pathogen growth. Gut microbial profiles are dependent on populations, geographical locations, and long-term dietary patterns resulting in individual uniqueness. Gut microbiota can be classified into enterotypes based on their patterns. Understanding gut enterotype enables us to interpret the capability in macronutrient digestion, essential substance production, and microbial co-occurrence. However, there is still no detailed characterization of gut microbiota enterotype in urban Thai people. In this study, we characterized the gut microbiota of urban Thai individuals by amplicon sequencing and classified their profiles into enterotypes, including *Prevotella* (EnP) and *Bacteroides* (EnB) enterotypes. Enterotypes were associated with lifestyle, dietary habits, bacterial diversity, differential taxa, and microbial pathways. Microbe–microbe interactions have been studied via co-occurrence networks. EnP had lower α-diversities than those in EnB. A correlation analysis revealed that the *Prevotella* genus, the predominant taxa of EnP, has a negative correlation with α-diversities. Microbial function enrichment analysis revealed that the biosynthesis pathways of B vitamins and fatty acids were significantly enriched in EnP and EnB, respectively. Interestingly, Ruminococcaceae, resistant starch degraders, were the hubs of both enterotypes, and strongly correlated with microbial diversity, suggesting that traditional Thai food, consisting of rice and vegetables, might be the important drivers contributing to the gut microbiota uniqueness in urban Thai individuals. Overall findings revealed the biological uniqueness of gut enterotype in urban Thai people, which will be advantageous for developing gut microbiome-based diagnostic tools.

## 1. Introduction

A hundred trillion microorganisms inhabiting the gastrointestinal tract play pivotal roles in maintaining and improving human health, including the utilization of indigestible nutrients [1], production of essential metabolites, e.g., short-chain fatty acids (SCFAs) [2,3], improvement of the immune system [4], reduction of inflammation [4], and elimination of toxins and pathogens [5,6,7]. Throughout the lifespan of a human, several factors, including mode of birth, breastfeeding, heredity, pathogenic infections, surgical procedures, antibiotic use, long-term dietary habits, behavioral changes, and geographical locations, can affect the composition and structure of these gut microbial communities [8,9,10,11]. The alteration of the microbial communities can lead to an imbalanced ecosystem known as dysbiosis. This can result in either an increase in harmful microorganisms or a decrease in beneficial bacteria and microbial diversity, raising the risk of various diseases [3,12]. Gut dysbiosis is known to be the starting point of various diseases. Therefore, understanding the normal state of gut microbiota is important to interpret health status.

The gut microbiota is personalized and varies widely among people, even in the same individual at different periods of time; nevertheless, the communities of the human gut microbiota are generally dominated by a few bacterial taxa [13]. In 2011, the enterotyping method was introduced to classify individuals based on common patterns and was named after the main feature of the microbiome profile [14]. Subsequently, extensive studies have revealed *Bacteroides*, *Prevotella*, and *Ruminococcus* as the three major enterotypes [14,15,16,17,18,19,20,21,22,23,24,25,26,27,28,29,30,31,32,33]. Enterotypes of the gut microbiota are relatively constant and strongly associated with repetitive dietary patterns. In other words, individuals who prefer a meat-rich diet belong to the *Bacteroides*-dominant enterotype, a high-carb diet is associated with the *Prevotella*-dominant enterotype, and high fruit or vegetable diet is relevant to the *Ruminococcus*-dominant enterotype [14,34,35,36,37,38]. Notably, biotin (vitamin B7), thiamine (vitamin B1), and heme biosynthetic pathways were discovered to be enriched in *Bacteroides*, *Prevotella*, and *Ruminococcus* dominant enterotypes, respectively [14]. These enriched synthetic pathways indicate the biochemical activities of the enterotype including nutrient synthesis, modulation of immunity, antibiotic production, and others. Understanding enterotypes and their characteristics can help us develop early diagnostic tools for disease prevention and personalized nutrition based on the gut microbiota profile. Although the predominant genera in the gut microbial community and the main functions of each enterotype have been discovered in healthy and dysbiotic individuals in many countries, the enterotypes and their characteristics in Thai individuals have not yet been well defined in detail. In addition, gut microbiota characteristics have also been influenced by ethnicity [39,40,41].

However, within the same ethnicity, people from distinct regions, especially between urban and rural areas, have different dietary patterns. Urban people have a tendency to consume western diets, which are high in meats, processed meats, refined grains, etc., rather than rural people [42]. Moreover, the prevalence of metabolic syndrome has also increased in the urban population [43,44,45]. Understanding the relationship between gut microbiota and health parameters in urban people may lead to guideline developments for disease prevention. In this study, we characterized the composition of the gut microbiota in 96 Thai urban individuals using fecal 16S rDNA amplicon-based metagenomics sequencing. The enterotyping method was used to stratify the profiles of the gut microbiota into enterotypes. Microbial co-occurrence networks were established for both enterotypes using Spearman’s correlation coefficient. Associations between the gut microbiota and host lifestyle parameters were statistically tested to determine specific features in a given enterotype. We compared enterotypes in various aspects of microbial markers, including diversity, host-associated lifestyle, enriched predicted functions, and co-occurrence networks leading to enhancing health based on the characteristics of enterotypes.

## 2. Materials and Methods

### 2.1. Study Design and Cohort

Two hundred urban volunteers who were living in Bangkok, which is the capital city of Thailand, and its vicinity, were recruited by the Systems Biology and Bioinformatics Unit, Pilot Plant Development and Training Institute, KMUTT in 2018. Volunteers were divided into 5 different age groups: 18–25, 26–35, 36–45, 46–55, and over 55 years old. Volunteers were asked to (i) sign an informed consent, (ii) complete questionnaires, (iii) collect their stool samples, and (iv) return their stool samples to the laboratory. The questionnaires consisted of sociodemographic and lifestyle factors, including health profile, e.g., gender, age, body mass index, delivery procedure, antibiotic use, smoking, alcohol consumption, allergies, type of diet, and probiotic use (see Appendix A for the questionnaire). Of the 200 participants, 131 participants returned stool samples to the laboratory and signed informed consent, while 69 participants who did not complete the whole procedure were excluded. In this study, 96 out of 131 samples were selected based on DNA quality.

### 2.2. DNA Extraction, Library Preparation, and 16S rDNA Gene Sequencing

DNA samples were extracted from stool samples using the QIAamp Fast DNA Stool Mini Kit (Qiagen, Hilden, Germany) based on enzymatic and mechanical lysis of cells. DNA samples were qualified using the NanoPhotometer^®^ N60/N50 spectrophotometer. The criteria were a DNA concentration of at least 10 ng/µL, OD 260/280 of 1.8–2.0, and OD 260/230 of 2.0–2.2. The hypervariable V4 regions of the 16S rDNA genes were amplified from the DNA samples using four sets of adapters and primers 515F and 806R (Appendix A). DNA libraries were prepared according to the MiSeq Reagent Kit V2 protocol and sequenced with 250 bp paired-end reads on the Illumina MiSeq platform at the Quebec Genome Institute, Canada.

### 2.3. Microbiome Data Analysis

Raw data quality was assessed using FastQC (v0.11.8) and MultiQC (v1.7) [46,47]. QIIME2 (v2019.7) was used to perform the microbiome analysis. Briefly, the adaptor and any preceding bases were trimmed at the 5’ end of the reads using the q2-cutadapt plugin [48]. Reads were truncated at the position where quality scores were reduced by less than 30. Paired-end reads were merged with an overlap of 25 nucleotides between forward and reverse reads, and chimeric sequences were filtered out using the q2-dada2 plugin [49]. Sequences were dereplicated into amplicon sequence variants (ASVs) using the q2-dada2 plugin, resulting in a feature table, representative sequences, and frequency of remaining sequences in each step [49]. The V4 hypervariable regions were extracted from 16S rDNA full-length sequences of the SILVA v132 SSURef NR99 using 515F and 806R primers. The taxonomic classifier was trained using the V4 sequences based on the naive Bayes classifier model using the q2-feature-classifier plugin [50,51]. ASVs were assigned for taxonomies based on the sklearn method using the q2-feature-classifier plugin. Unassigned, mitochondria, and chloroplast sequences were removed using the q2-taxa plugin. The feature table containing absolute abundances of samples and taxa was normalized using a rarefying method with a sequencing depth of 25,000 reads per sample and 5000 bootstrapping iterations to improve the reproducibility of the data using the q2-feature-table plugin.

### 2.4. Enterotyping

Enterotyping was used to classify samples into clusters based on the similarity of microbiome profiles [14]. The taxonomic abundance table at the genus level normalized by the rarefying method was used as input for enterotyping. The square root of Jensen–Shannon divergence (JSD) was used to measure dissimilarity between communities. The Calinski–Harabasz index (CH), the ratio of the sum of squares of dissimilarity between clusters to the sum of squares of dissimilarity within clusters, was calculated to determine the optimal number of clusters.

The silhouette coefficient was used as an indicator to calculate the goodness of the clustering technique from −1 (data were assigned to the wrong group) to 1 (clusters were separated) [52,53,54]. The clustering algorithm, partitioning around medoids (PAM), a supervised method, was used to group the taxonomic profiles according to the estimated number of clusters. Dissimilarities between communities were examined and visualized using PCoA, and the clustering results were overlaid on the graph.

### 2.5. Statistical Analysis

Kolmogorov–Smirnov tests were conducted to test the normality of the lifestyle-related data. The independence of lifestyle-related data between groups was examined using Fisher’s exact test. Differences in the relative abundance of each genus between enterotypes were examined using the Mann–Whitney U test. Chi-square tests with a cutoff *p*-value of less than 0.05 were performed to examine the association between lifestyle parameters and enterotypes.

### 2.6. Microbial Diversity Analysis

Alpha diversity was calculated at the genus level using Simpson’s diversity, Shannon’s diversity, Chao1 richness, and Pielou’s evenness indices via the package vegan (v2.4.2) and visualized using the package ggplot2 implemented in the R environment (v3.6.2). The difference in alpha diversity between two enterotypes was statistically tested using the Mann–Whitney U test with a cutoff *p*-value of less than 0.05. Correlation between alpha diversity indices and microbial abundance was performed using Spearman’s correlation in the SPSS program (v26) with a cutoff *p*-value of less than 0.05. Beta diversity was used to cluster the samples according to their microbiome profiles using PCoA based on Bray–Curtis distances [55]. PCoA plots were then overlaid with metadata from the questionnaires to classify groups of participants using the QIIME2 EMPeror plugins [56]. PERMANOVA tests were employed to assess whether the microbial communities in each lifestyle parameter are different with a cutoff *p*-value < 0.05 using the vegan (v2.5.7) and phyloseq (v1.36.0) packages in the R environment (v4.1.1).

### 2.7. Differential Abundance Analysis

The description of the experimental linear discriminant analysis effect size (LEfSe) on the Galaxy v1.0 platform was used to detect significantly differential abundant features between enterotypes at the genus level, where the linear discriminant analysis (LDA) criterion had a value of 2.0 or greater [57].

### 2.8. Co-Occurrence Network Construction and Network Property Measurement

In each enterotype, microbe–microbe interactions were determined by Spearman’s rank correlation coefficient [58] and filtered by the cutoff criteria of $\left|\rho \right|$ ≥ 0.5, and *p*-value < 0.05. Microbe–microbe interactions that met the criteria were used to create a co-occurrence network in which nodes and edges represent microbes and interactions, respectively. Co-occurrence networks were evaluated using several network property parameters, including the number of nodes, number of edges, hubs, degree, clustering coefficient, network diameter, the average number of neighbors, network density, network heterogeneity, and network centralization using CytoCluster v2.1.0 [59] and NetworkAnalyzer v4.4.6 [60] packages in Cytoscape v3.8.0 [61]. Hierarchical clustering analysis based on complete-linkage was performed to cluster taxa similarity using stats (v4.1.1), Hmisc (v4.5.0), and NetCluster (v0.2) [62] packages in the R environment (v4.1.1).

### 2.9. Microbial Function Prediction and Pathway Enrichment Analysis

Phylogenetic Investigation of Communities by Reconstruction of Unobserved States (PICRUSt2 v2.3.0b0) [63] was applied to predict microbial metabolic functions including KEGG pathways, KEGG orthologs, and EC numbers. Pathway and gene abundances were estimated for each sample, based on the copy numbers of 16S rDNA genes of microbes found in the sample [64]. Statistical Analysis of Taxonomic and Functional Profiles (STAMP v2.1.3) was then employed to test whether functional abundances differed significantly between enterotypes using Welch’s *t*-test for unequal variance. The Benjamin–Hochberg (BH) correction for multiple hypotheses was used to estimate the false discovery rate (FDR). Functions were found to be significantly different between enterotypes when the q-value was less than 0.05 and the absolute value of the effect size was greater than 0.1 [65,66]. The value of the mean proportion indicates which functions are enriched in which enterotype.

## 3. Results

### 3.1. Demographic Data of Enrolled Participants from the MODGUT Project

The ninety-six participants who joined this project were divided into 5 age groups: 20 individuals aged 18–25 years, 29 individuals aged 26–35 years, 31 individuals aged 36–45 years, 6 individuals aged 46–55 years, 4 individuals aged over 55 years, and 6 individuals with no recorded data. In total, 61, 29, and 6 individuals were female, male, and unknown gender, respectively. The mean age of all participants was 34.17 ± 10.57 years. Participants were classified into 4 categories based on body mass index (BMI): 9 underweight (less than 18.5), 48 normal weight (18.5–24.9), 23 overweight (25.0–29.9), 10 obese (at least 30.0), and 6 unknown data. The mean BMI was 22.58 ± 3.68 kg/m^2^. A total of 62 of the 96 individuals had normal dietary patterns, 27 did not consume raw foods, and only one person had a vegan diet (Table 1). Further metadata were shown in Appendix A.

### 3.2. Thai Gut Microbiota Profiling

The average number of raw paired-end reads per sample was 68,867, and the total number of raw reads was 6,611,209. After preprocessing, a total of 3,978,787 reads remained, and the average number of reads was 41,446 per sample (Appendix A). Ninety-one percent of all sequences had a length of 253 nucleotides, which corresponds to the theoretical length of the hypervariable V4 region [67] as shown in Appendix A. In total, 14 phyla, 22 classes, 35 orders, 79 families, and 271 genera were discovered in this dataset. The relative abundance of these taxa might be used as a reference range for Thai people, which could be further applied in health status prediction (Appendix A). Characterization of the gut microbiota of Thai individuals revealed that Bacteroidetes (54.64 ± 19.01%) and Firmicutes (37.26 ± 17.20%) were mainly found in all samples. In addition, phyla Proteobacteria (5.67 ± 6.11%), Fusobacteria (0.85 ± 3.38%), Verrucomicrobia (0.51 ± 1.83%), Actinobacteria (0.50 ± 0.95%), and Elusimicrobia (0.12 ± 1.15%) were also detected in this study. *Bacteroides*, *Prevotella*, and *Faecalibacterium* were identified as the first three predominant genera. Eighty-seven genera had a relative abundance greater than 1%. Interestingly, about 90% of this cohort occupied Actinobacteria and about 80 percent of them had the genus *Bifidobacterium* in their intestinal tract. The alpha diversity of the gut microbiota based on Shannon’s diversity index ranged from 0.373 to 3.435, Simpson’s diversity index ranged from 0.118 to 0.943, Pielou’s evenness index ranged from 0.112 to 0.734, and the Chao1 richness index ranged from 22 to 109. Statistical tests revealed that there was no significant difference in alpha diversities including Chao1, Pielou, Simpson, and Shannon indices in age group, BMI category, and gender (Appendix A). The beta diversity of the gut microbiota profiles based on the Bray–Curtis’s distance was calculated and ordinated by PCoA (Appendix A). PERMANOVA revealed that gut microbiota profiles were not significantly associated with any host lifestyle parameters, e.g., age interval, BMI category, and gender (Appendix A). However, Fisher’s exact test revealed that there were a few highly abundant taxa (at least 1% abundance) significantly associated with age interval, gender, and BMI category (Appendix A).

### 3.3. Association of Microbes and Diversity in Urban Thai Subjects

To investigate the correlation between microbes and the diversity of the microbiome, we calculated Spearman’s correlation coefficient between the microbial abundances in the samples and the diversity indices (Appendix A). Forty-nine out of 87 taxa having at least 1% relative abundance were significantly correlated with at least one diversity index with the cutoff criteria ($\left|\rho \right|$ ≥ 0.35, *p*-value < 0.05). *Bacteroides* was negatively correlated with the Chao1 richness index (ρ = −0.234, *p*-value = 0.022). Similarly, *Prevotella 9* was negatively correlated with Shannon’s diversity index (ρ = −0.316, *p*-value = 1.73 × 10^−3^), Simpson’s diversity index (ρ = −0.336, *p*-value = 8.20 × 10^−4^), and Pielou’s evenness index (ρ = −0.373, *p*-value = 1.85 × 10^−4^). This implies that an increased amount of these predominant bacteria may lead to lower microbial diversity in the gut. Interestingly, microbes from the Ruminococcaceae family, e.g., Ruminococcaceae NK4A214 group, Ruminococcaceae UCG-002, Ruminococcaceae UCG-003, and Ruminocaccaceae UCG-005 had strong positive correlations with all α-diversity indices with criteria ρ ≥ 0.5 and *p*-value < 0.05 (Appendix A), suggesting that a higher proportion of Ruminococcaceae had a high association with the increase of α-diversity. On the other hand, *Fusobacterium* has negative correlations with all diversity indices (Shannon, ρ = −0.306, *p*-value = 0.002; Simpson, ρ = −0.226, *p*-value = 0.027; Chao1, ρ = −0.493, *p*-value = 3.40 × 10^−7^; Pielou, ρ = −0.237, *p*-value = 0.02).

### 3.4. Stratification and Characterization of Thai Individuals Based on Their Gut Microbiota Profiles

Thai individuals were stratified into two enterotypes based on their gut microbiota composition: *Prevotella*- and *Bacteroides*-dominant enterotypes hereafter referred to as EnP and EnB, respectively (Figure 1A). Two-thirds of the participants belonged to enterotype B. More than half (58.90% of the average relative abundance) of total abundance in EnP was occupied by the genus *Prevotella 9* (Figure 1B), while *Bacteroides* was the most dominant genus in EnB with an average relative abundance of 37.80% (Figure 1C). According to the result of differential abundance analysis between two enterotypes by LEfSe, 64 taxa and 24 taxa were significantly enriched in EnB and EnP, respectively (Figure 2). For example, *Prevotella 9*, *Elusimicrobium*, *Anaerovibrio*, *Succinivibrio*, *Prevotella 2*, *Alloprevotella*, and *Megasphera* genera were significantly enriched in EnP, while *Bacteroides*, *Faecalibacterium*, *Phascolarctobacterium*, *Sutterella*, *Alistipes*, *Lachnoclostridium*, *Parabacteroides*, *Roseburia*, *Streptococcus*, *Blautia*, *Bilophila*, *Fusicaberibacter*, *Flavonibacter*, *Veillonella*, *Collinsella*, *Anaerostipes*, *Butyricicoccus*, *Tyzzerella*, and *Hungatella* significantly predominated in EnB. Notably, *Fusobacterium*, which was reported to be associated with gut dysbiosis and intestinal inflammation possibly leading to the development of colorectal cancer and ulcerative colitis [68,69], was found in 12 individuals with at least 1% relative abundance. Eleven out of twelve participants belonged to EnB, whereas the remaining one belonged to EnP. Fisher’s exact test also showed the association between *Fusobacterium* and EnB (*p*-value = 0.056). In addition, *Roseburia*, the butyrate-producing bacteria, was detected in 19 participants with at least 1% relative abundance. Fisher’s exact test indicated that *Roseburia* was significantly enriched in EnB at the *p*-value of 0.003 (18 out of 19 participants belonged to EnB whereas one belonged to enterotype P). On the other hand, 11 participants had at least 1% relative abundance of *Bifidobacterium*, a well-known beneficial microbe, but only one belonged to EnP (*p*-value = 0.093) (Appendix A).

Diversity analysis between EnB and EnP using the Mann–Whitney U test shows that EnB had significantly higher α-diversity than EnP based on Pielou’s evenness index (*p*-value = 1.37 × 10^−5^), Simpson’s diversity index (*p*-value = 8.26 × 10^−5^), and Shannon’s diversity index (*p*-value = 1.82 × 10^−4^), while the Chao1 richness index (*p*-value = 0.71) was not significantly different between both enterotypes (Figure 3). Moreover, the gut microbiota patterns of both enterotypes were significantly distinct according to PERMANOVA (*p*-value = 1 × 10^−4^).

To investigate the association between enterotypes and lifestyle parameters, e.g., age, BMI, frequency of exercise, and hours of sleep, Fisher’s exact tests indicate that the lifestyle variables including consumption of eggs and meat (*p*-value = 0.060) and poultry (*p*-value = 0.040) were statistically different between enterotypes (Appendix A). In addition, the BMIs of EnP were slightly higher than those of EnB (Table 1 and Appendix A).

### 3.5. Microbial Co-Occurrence Network and Clusters of Strongly Related Microbes

The microbial co-occurrence network describes microbe–microbe relationships, where nodes and edges represent microbial taxa and their interactions, respectively. 87 out of 271 taxa having at least 1% of relative abundance were included. Microbe–microbe correlations were calculated using Spearman’s correlation coefficient. From the correlations of microbiome profiles that met the criteria (*p*-value < 0.05 and $\left|\rho \right|$ ≥ 0.5), the separated co-occurrence networks of EnP and EnB were constructed, yielding 67 and 42 nodes and 181 and 109 edges, respectively (Figure 4). Based on taxonomic assignment, the majority of microbes in both networks (39 nodes in *p* and 29 nodes in B) belonged to the phylum Firmicutes (Appendix A). Remarkably, *Prevotella* in EnP had negative interactions with 11 other taxa in EnP (Appendix A). In contrast, *Bacteroides* in EnB had only one negative interaction with Ruminococcaceae UCG-002 (Appendix A).

Subsequently, network properties including the number of nodes, number of edges, node degree, clustering coefficient, network diameter, the average number of neighbors, network density, network heterogeneity, and network centralization were measured to compare the complexity and robustness of the two ecosystems (Table 2). The nodes with the 10% highest degrees were identified as hubs, including, *Alistipes* (13 degrees), *Butyricimonas* (15 degrees), *Lachnospiraceae UCG-010* (13 degrees), *Oscillibacter* (17 degrees), Ruminococcaceae NK4A214 group (17 degrees), Ruminococcaceae UCG-002 (16 degrees), Ruminococcaceae UCG-005 (21 degrees), and Ruminococcus torques group (15 degrees) in EnP, and *Christensenellaceae R-7 group* (13 degrees), *Eubacterium coprostanoligenes group* (17 degrees), Ruminococcaceae NK4A214 group (16 degrees), Ruminococcaceae UCG-002 (18 degrees), Ruminococcaceae UCG-005 (13 degrees), and Ruminococcaceae UCG-010 (13 degrees) in EnB. The network of EnP was obviously larger than that of EnB in terms of the number of nodes, edges, and network diameters, suggesting that microbes in EnP have more microbial variants than those of EnB to maintain ecological stability. The clustering coefficient, which indicates how any three nodes interact with each other [70], was higher in network B, suggesting that EnB has more connectivity than EnP. Both networks had an average number of neighbors that was approximately equal to 6. The network density, which approaches zero and mainly describes the density of the edges [71,72], suggests that microorganisms generally live in the form of clusters. Both communities had network centralization approaching zero, implying that the microorganisms live together as a module, with some microbes responsible for the hub as identified previously.

Based on hubs, network density, and centrality, microorganisms generally live as an interactive community in an ecosystem. To understand the importance of microbial co-occurrence modules, microbial profiles were clustered using the hierarchical clustering method with complete linkage in each enterotype (Appendix A), resulting in 4 and 6 highly correlated clusters for EnP (Appendix A) and EnB (Appendix A), respectively (Appendix A).

### 3.6. Metabolic Potential Prediction of Prevotella and Bacteroides Enterotypes

Metabolic pathway prediction of microbial communities was performed using PICRUSt [63] on 153 metabolic pathway profiles, which were annotated for the gut microbiota of all participants based on the KEGG database. The predicted abundance of metabolic pathways was evaluated for the analysis of the difference in abundance between both enterotypes. Thirty-eight out of 153 metabolic pathways differed significantly between both enterotypes (Figure 5), with 26 and 12 metabolic pathways enriched in EnP and EnB, respectively. Interestingly, several of these pathways were associated with the production of B vitamins, including thiamine (B1) metabolism (KO00730, q-value = 3.33 × 10^−8^), riboflavin (B2) metabolism (KO00740, q-value = 1.95 × 10^−13^), pyridoxine (B6) metabolism (KO00750, q-value = 1.57 × 10^−9^), nicotinate (B3) and nicotinamide metabolism (KO00760, q-value = 1.04 × 10^−10^), pantothenate (B5) and CoA biosynthesis (KO00770, q-value = 1.09 × 10^−10^), and folate (B9) biosynthesis pathway (KO00790, q-value = 3.97 × 10^−6^). In addition, amino acid biosynthetic pathways were also enriched in enterotype P, i.e., alanine, aspartate, and glutamate metabolism (KO00250, q-value = 5.06 × 10^−7^); cysteine and methionine metabolism (KO00270, q-value = 8.69 × 10^−13^); valine, leucine, and isoleucine biosynthesis (KO00290, q-value = 3.42 × 10^−6^); phenylalanine, tyrosine, and tryptophan biosynthesis (KO00400, q-value = 1.58 × 10^−9^); D-glutamine and D-glutamate metabolism (KO00471, q-value = 1.70 × 10^−14^); and D-alanine metabolism (KO00473, q-value = 2.34 × 10^−5^). However, ansamycin biosynthesis (KO01051), which is responsible for antibiotic production, was significantly enriched in EnB (q-value = 1.99 × 10^−3^). Secondary bile acid biosynthesis (KO00121) was another metabolic pathway that was significantly enriched in EnB (q-value = 7.34 × 10^−13^). Biotin (vitamin B7) metabolism (KO00780), already known to be relevant to carbohydrate utilization and to reduce inflammation of human dendritic cells [73], was also significantly overrepresented in EnB (q-value = 9.10 × 10^−9^). Furthermore, PICRUSt2 indicates that SCFA-producing bacteria including acetate, butyrate, and propionate presented in both enterotypes. STAMP shows that the bacterial abundances of these SCFA-related pathways were not significantly different between enterotypes.

## 4. Discussion

The profiles of the gut microbiota are personalized relying on various factors, but they often share common patterns that are co-existing with the same microbes with similar relative abundance, enabling us to cluster these profiles into groups. Enterotyping, a clustering method based on Jansen–Shannon distance, was introduced to partition microbiome profiles based on their general patterns. The clusters were then named after the predominant genus normally found in the imbalanced gut microbiota [74]. *Prevotella* enterotype (EnP) and *Bacteroides* enterotype (EnB) have been detected in the human gut microbiota of different populations [14,15,16,17,18,19,20,21,22,23,24,25,26,27,28,29,30,31,32,33]. In this study, one-third of all participants belonged to EnP, while the remaining two-thirds were corresponding with EnB, respectively. This is consistent with other studies that reported that gut microbiota of urban people occupied *Bacteroides* rather than *Prevotella* [32,75,76]. Other enterotypes could not be detected in our dataset. This could be due to the limited sample size or because the individuals were not distinct groups [33]. Moreover, it might depend on traditional dietary patterns, genetics, and lifestyle habits. *Ruminococcus* and other enterotypes were identified in other studies and the *Ruminococcus* enterotype was found in some populations [14,19,20,21,22,23,24,25,26,27,28,29,30,31,32,33]: *Blautia* enterotype was predominant in Japanese people [29]; *Bifidobacterium* enterotype in Thai and Dutch children [31,32]; *Escherichia* enterotype in Taiwanese people [21,28]; *Ruminococcus-Bifidobacterium* enterotype in Chinese people [14,33]; and *Prevotella-Bacteroides* enterotype in Americans [27]. In order to understand the characteristics and roles of gut enterotypes derived from this cohort, we investigated the relationship between enterotypes and microbiome diversity. Interestingly, we observed significantly higher α-diversity of gut microbiota communities in EnB than in EnP, which was consistent with a previous study [21] but also in contrast to some previous reports [31,32]. Moreover, the *Prevotella* to *Bacteroides* (P/B) ratio, another health indicator used to determine the ability to reduce weight on a specific food, was completely different between the EnP (59.85 ± 122.86) and the EnB (0.06 ± 0.21) [74,77,78]. Therefore, the P/B ratio can be used as a marker for the separation of EnP and EnB. EnP was observed to have a slightly higher BMI than EnB, which is consistent with the previous report [74]. Specifically, we found that EnB individuals consumed significantly more eggs, meat, and poultry than those of EnP, as revealed by statistical tests. Similarly, western individuals, who are predominantly of the *Bacteroides*-based enterotype, have a relatively low P/B ratio because the western diet is rich in red meat from terrestrial food animals. These results are consistent with previous findings suggesting that the *Bacteroides* genus may be increased in the intestinal tract by the consumption of a meat-rich diet, while *Prevotella* may be enriched by a high-fiber diet [34,36,37]. The discovery and characterization of enterotypes could bring benefits to the healthcare system.

Differential abundance analysis based on LEfSe revealed the microbial signature of each enterotype. Remarkably, the number of significantly enriched taxa in EnB was clearly greater than that in EnP. This can be explained that the α-diversity of EnB was significantly higher than that of EnP. In the EnB, the genus *Ruminococcus*, one of the resistant starch degraders as well as producers of SCFAs and natural antibiotics responsible for improving and maintaining the human gut symbiosis [79,80,81,82,83,84], was significantly predominant. Moreover, several members of the Ruminococcaceae family were positively correlated with species richness and evenness (Appendix A), which are among the important indices of human health [85,86], suggesting that microbes from the Ruminococcaceae family might provide food sources for other microbes leading to an increase of microbial diversity in the intestinal tract, leading to better health. In general, there are not only beneficial microbes in the human gut but also harmful microbes at a very low level. *Elusimicrobium*, which normally decreases in patients with type 2 diabetes, was enriched in EnP [87,88]. However, harmful microbes at low levels do not damage the host because their composition is regulated by other beneficial and commensal microbes. An inappropriate lifestyle, such as a high-fat diet, can affect the composition of the gut microbiota and lead to an imbalance (or gut dysbiosis), so that commensal bacteria under certain circumstances become pathobionts, which can harm the host, and some harmful or pathogenic bacteria may increase and lead to pathogenesis [89,90]. In this study, these harmful and pathogenic bacteria, including *Collinsella*, *Fusobacterium*, *Odoribacter*, *Peptococcus*, *Tyzzerella*, and *Veillonella* with low relative abundance, were identified and the association with enterotypes was investigated. *Odoribacter* was significantly enriched in EnB (Figure 2), which has previously been reported to be overrepresented in neurological-related cases, such as attention deficit hyperactivity disorder (ADHD) [91] and depression [92]. *Collinsella* was discovered in EnB (Figure 2), which was strongly correlated with rheumatoid arthritis [93]. *Veillonella*, known to cause liver cirrhosis, was significantly enriched in EnB [94]. Interestingly, *Tyzzerella* was significantly increased in EnB (Figure 2), which is consistent with the previous study which reported that *Tyzzerella* was enriched in individuals at high risk for cardiovascular disease [95] and rheumatoid arthritis [96]. In addition, Fusobacteria associated with *Fusobacterium spp.* bacteremia, colorectal cancer, periodontitis, and ulcerative colitis, were detected with relatively higher abundance in EnB than in EnP [68,69,89,97,98,99,100,101]. Overall, although a number of beneficial bacteria were enriched in EnB rather than in EnP, some harmful bacteria were also overrepresented in EnB. The results indicate that urban people are likely to occupy a higher abundance of harmful bacteria leading to an increased risk of several diseases, possibly due to the alteration of dietary patterns. Therefore, a balance between *Prevotella* and *Bacteroides* in the gut microbiota is preferable for maintaining the abundance of beneficial and harmful bacteria, improving the digesting ability of various food types, and preserving gut microbial diversity.

Functional enrichment analysis was examined to infer enterotype-specific microbial pathways based on PICRUSt2 and STAMP. Overall, the number of microbial pathways in the amino acid metabolism, the nucleotide metabolism, and the cofactor and vitamin metabolism were enriched in EnP more than those in EnB. On the other hand, carbohydrate and lipid metabolic pathways were overrepresented in EnB more than in those of EnP. The pathways for B vitamin metabolism, i.e., B1, B2, B3, B5, B6, and B9, and the pathways for essential amino acids metabolism, i.e., valine, leucine, isoleucine, phenylalanine, and tryptophan, were enriched in EnP compared to EnB (Figure 5). B vitamin metabolic pathways have been reported to be associated with the genus *Prevotella* in the previous study [102]. EnB may have the potential to protect against mycobacterial infections due to a significant enrichment of ansamycin biosynthesis (KO01051, q-value = 1.99 × 10^−3^), which produces antibiotics against the mycobacteria [103]. In addition, the glycosaminoglycan (GAG) degradation pathway, which is responsible for maintaining water in tissues and preventing skin aging, was significantly enriched in EnB (KO00531, q-value = 9.86 × 10^−14^) [104]. Interestingly, the secondary bile acid (SBA) pathway was significantly enriched in EnB compared to EnP (q-value = 7.34 × 10^−13^). SBAs, consisting of deoxycholic acid and lithocholic acid, are normally absorbed in the colon and flowed in the enterohepatic circulation [105,106,107]. These metabolites can be increased by high-fat diet intakes, possibly leading to various conditions, including colonic inflammation, colon cancer, and inflammatory bowel disease [105,106,108,109,110,111]. However, due to the limitation of amplicon-based sequencing, these predicted pathways could be an inference to the role of enterotypes. Therefore, a metagenomic approach is required for the functional identification of enterotypes.

Microbes living in a community normally communicate with each other to produce and transfer metabolites between themselves and their host [112]. Constructing a microbiome network (or co-occurrence network) in which nodes and edges represent microbes and their interactions may be a promising way to understand how these microbes communicate and interact with the host. To understand the behavior of bacterial communities, both enterotype networks were constructed and their network properties, including network size, clustering coefficient, number of nodes, number of edges, node degree, average correlation, the average number of neighbors, network density, network heterogeneity, and network centralization were evaluated [113]. Based on the network properties, network P had more microbial members and interactions than network B, while network B had greater robustness than network P, as inferred from the clustering coefficient and the average number of neighbors (Table 2). An individual with higher microbial network diversity and robustness could possess more diverse microbial functions, including producing nutrients and vitamins, enhancing host immunity, and decreasing inflammation [114,115]. All hubs, bacteria having high node degree, discovered in this study, including Ruminococcaceae NK4A214 group, Ruminococcaceae UCG-002, Ruminococcaceae UCG-005 from both enterotypes, and Ruminococcaceae UCG-010 in EnB, belonged to the same family Ruminococcaceae, which is a resistant starch and dietary fiber decomposer leading to produce SCFAs, and feeding to other microbes in the same ecosystem [81,116,117,118]. The main staple food in Thailand is rice and vegetables, although animal product consumption is on the rise [119]. Westernized dietary habits, consisting of a high-fat, meat-rich, and high-sugar diet, might induce steatohepatitis [120]. It has been previously reported that *Ruminococcus* abundance is reduced in inflammatory bowel disease [121]. This suggests that Ruminococcaceae may be essential bacteria for all humans and contribute to the stability of the gut ecosystem [122,123]. In addition, the genus *Fusobacterium*, which has been associated with various diseases, e.g., colorectal cancer [68], *Fusobacterium* spp. bacteremia [89], acute pharyngitis [124], and periodontitis [125], was negatively correlated with several Ruminococcaceae in both enterotypes (Figure 4), implying that the presence of Ruminococcaceae may help reduce the abundance of harmful *Fusobacterium.* In the network P, *Prevotella* had only negative interactions with other bacteria (Appendix A). Furthermore, the relative abundance of *Prevotella* in EnP was negatively correlated with the species evenness (Appendix A), indicating that the presence of *Prevotella* led to a decrease in microbial diversity. The co-occurrence network of gut microbiota can provide us with an understanding of microbe–microbe and microbe–host interactions through the enterotypes, facilitating further applications regarding gut microbiota intervention and early screening diagnostic tools.

## 5. Conclusions

Enterotypes of Thai urban individuals were identified and characterized based on microbial profiles, host lifestyle, and dietary habits to understand the health benefits and the disease risks. A comparison of gut enterotypes in Thai urban people revealed higher diversity in EnB, higher abundance of beneficial and harmful bacteria in EnB, enriched B vitamin and amino acid biosynthesis in EnP, enterotype-specific differential abundant taxa, and unique microbial communities. Ruminococcaceae, the pivotally important microbes in a Thai cohort, might contribute to the stability of the gut ecosystem by degrading cellulose, cross-feeding, and cross-talking with SCFA producers. Nevertheless, further investigation into metabolomics and metagenomics is warranted for confirming microbial functions and microbe–microbe interactions. The overall results of this study may be useful to develop personalized nutrition for enhancing human health and preventing gut dysbiotic-driven disorders.

## Figures and Tables

**Figure 1 microorganisms-11-00136-f001:**
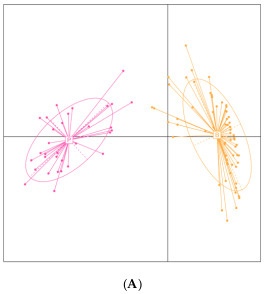
Gut microbiome enterotype and bacterial taxa in each enterotype: (**A**) gut microbiome profiles were separated into EnP and EnB based on the enterotyping method and were visualized by PCoA, in which *X*-axis and *Y*-axis represent the first and second principal components. Ten highest average taxonomic relative abundances at the genus level in EnP (**B**) and EnB (**C**) were demonstrated by boxplots.

**Figure 2 microorganisms-11-00136-f002:**
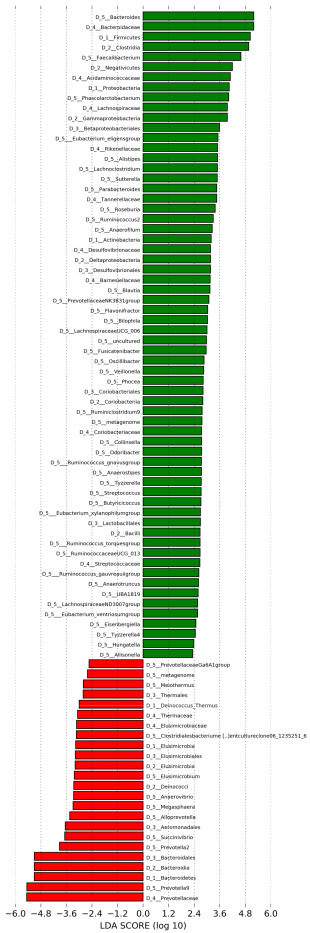
Differential abundance analysis. Differential abundant taxa were identified for EnP (red) and EnB (green) using LEfSe with the criteria LDA score ≥ 2.0 and *p*-value < 0.05.

**Figure 3 microorganisms-11-00136-f003:**
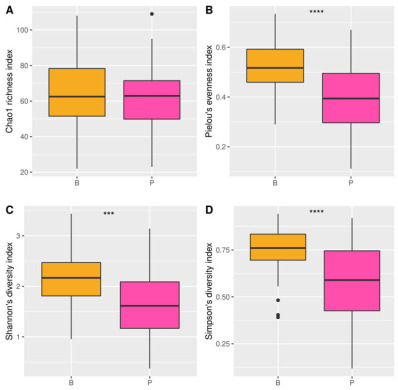
α-diversity analysis of gut microbiome composition in EnP and EnB with (**A**) Chao1, (**B**) Pielou’s evenness, (**C**) Shannon’s diversity, and (**D**) Simpson’s diversity indices. The symbols *** and **** represent *p*-value < 0.001 and *p*-value < 0.0001, respectively.

**Figure 4 microorganisms-11-00136-f004:**
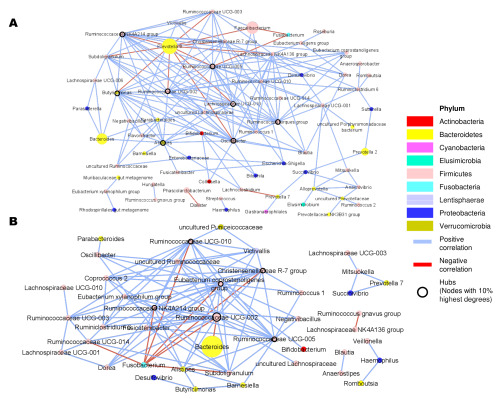
Co-occurrence network. Co-occurrence network of EnP (A) and EnB (B) were constructed based on Spearman’s correlation between microbial profiles with the criteria $\left|\rho \right|$ ≥ 0.5 and *p*-value < 0.05. Nodes and edges represent taxa and microbe–microbe relationships, and node and edge colors denote bacterial phylum and type of correlation, respectively.

**Figure 5 microorganisms-11-00136-f005:**
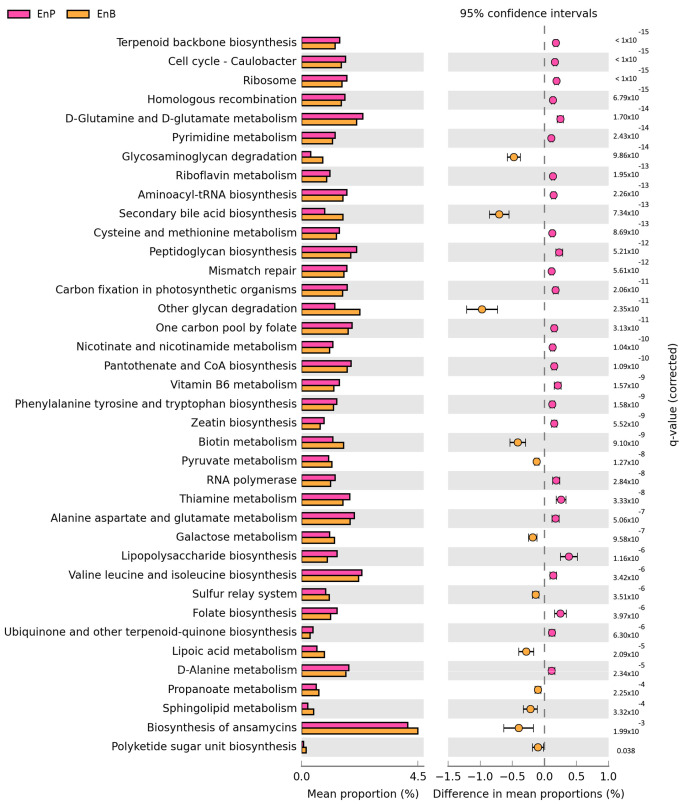
Enrichment analysis of microbial metabolic function. A comparison of predicted metabolic functions between enterotypes was performed by enrichment analysis. Bar plots display the mean proportion (%) of metabolic pathways for EnP and EnB. Differences in mean proportions were conducted using two-sided Welch’s *t*-test and multiple hypothesis correction by Benjamin’s test with the criteria of difference in mean proportion > 0.1% and q-value < 0.05. The enriched pathways in EnP and EnB were denoted by pink and orange dots, respectively.

**Table 1 microorganisms-11-00136-t001:** Characteristics of participants in each enterotype.

Lifestyle Variables	N (n = 96)	EnP (n = 32)	EnB (n = 64)	*p*-Value ^1^
Gender				0.339
Male	29	12	17	
Female	61	17	44	
Age (years)	34.17 ± 10.57	34.31 ± 10.75	34.10 ± 10.56	0.972
18 to 25	20	6	14	
26 to 35	29	10	19	
36 to 45	31	10	21	
46 to 55	6	2	4	
Over 55	4	1	3	
BMI ^2^ (kg/m^2^)	22.58 ± 3.68	22.78 ± 3.40	22.48 ± 3.82	0.884
Underweight	9	2	7	
Normal	48	16	32	
Overweight	23	8	15	
Obese	10	3	7	
Types of diets				0.586
Vegetable and animal meat	62	22	40	
Vegetable and animal meat but not consume raw meat	27	7	20	
Vegan	1	0	1	

^1^ *p*-values were derived from Fisher’s exact test for testing the difference of lifestyle variables between enterotypes. ^2^ BMI stands for Body Mass Index.

**Table 2 microorganisms-11-00136-t002:** Network properties of enterotypes.

Network Properties	EnP	EnB
Number of nodes	67	42
Number of edges	181 (Positive: 166, Negative: 15)	109 (Positive: 100, Negative: 9)
Clustering coefficient	0.353	0.485
Network diameter	7	5
Average number of neighbors	6.069	6.516
Network density	0.106	0.217
Network centralization	0.271	0.409

## Data Availability

Raw sequences of the hypervariable V4 16S rDNA region in FASTQ file format for all samples have been deposited in the ENA repository under the BioProject accession number PRJEB41122 with the sample ID from SAMEA7538457 to SAMEA7538552.

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
