# Peer review of "Characterization of the Gut Microbiota in Urban Thai Individuals Reveals Enterotype-Specific Signature"

_microorganisms, 2023, doi:10.3390/microorganisms11010136_

Round 1

Reviewer 1 Report

The paper “Characterization of the Gut Microbiota in Urban Thai Individuals Reveals Enterotype-Specific Signature” addresses a topic worthy of investigation; the experiments were generally well-designed, and data properly analyzed. However, I have some questions and suggestions to improve the paper:

1) Did authors test gender effect? Although female and male sub-samples were composed of a different number of individuals, paired comparisons or other approaches could be useful. Authors tested global diversity and did not find any difference; I suggest focusing on some target genera or taxa

2) I see that authors asked to participants their weight and so they had the classification as underweight, obese etc…; also, in this case global diversity could mask some affect at genera level.

My experience on gut microbiota suggests that a focus on some taxa rather than on the whole microbiome could offer many details.

Author Response

Thank you very much for your kind review. Your comments and suggestions are very useful for improving our manuscript. We carefully revised the manuscript according to your suggestions and comments. We prepared the responses and put them in the attachment. Please see the attachment. 

Reviewer 2 Report

The present study is very interesting, and represent a step further in the state-of-the-art of characterization of the gut microbiota in relation with factors such as  lifestyle, dietary habits, bacterial diversity, differential taxa, and microbial pathways, and ethnicity.

Minor aspects:

In section 2.1. Study design and cohort, there is no mention about the selection of Thai volunteers from urban, only later in results section. Please specify in section 2.1 as well, and explain why from urban? Later in the conclusion section it is highlighted again about the comparison of enterotypes in Thai urban individuals.

Do you think there is a significant difference between urban and rural? Maybe these aspect shall be discussed in introduction or discussions.

Literature is up to date.

Author Response

(The authors gave the same response as above.)

Round 2

Reviewer 1 Report

Authors addressed my issues; thus, the paper can be accepted for publication